# A Fluidics-Based Biosensor to Detect and Characterize Inhibition Patterns of Organophosphate to Acetylcholinesterase in Food Materials

**DOI:** 10.3390/mi12040397

**Published:** 2021-04-03

**Authors:** Dang Song Pham, Xuan Anh Nguyen, Paul Marsh, Sung Sik Chu, Michael P. H. Lau, Anh H. Nguyen, Hung Cao

**Affiliations:** 1Biomedical Engineering Department, University of California Irvine, Irvine, CA 92697, USA; dspham1@uci.edu (D.S.P.); axnguye2@uci.edu (X.A.N.); sungsc@uci.edu (S.S.C.); 2Electrical Engineering and Computer Science Department, University of California Irvine, Irvine, CA 92697, USA; marshp@uci.edu; 3Sensoriis, Inc., 7500 212th St SW Ste 208, Edmonds, WA 98026, USA; michaelphlau@comcast.net

**Keywords:** electrochemical biosensors, acetylcholinesterase, organophosphate

## Abstract

A chip-based electrochemical biosensor is developed herein for the detection of organophosphate (OP) in food materials. The principle of the sensing platform is based on the inhibition of dimethoate (DMT), a typical OP that specifically inhibits acetylcholinesterase (AChE) activity. Carbon nanotube-modified gold electrodes functionalized with polydiallyldimethylammonium chloride (PDDA) and oxidized nanocellulose (NC) were investigated for the sensing of OP, yielding high sensitivity. Compared with noncovalent adsorption and deposition in bovine serum albumin, bioconjugation with lysine side chain activation allowed the enzyme to be stable over three weeks at room temperature. The total amount of AChE was quantified, whose activity inhibition was highly linear with respect to DMT concentration. Increased incubation times and/or DMT concentration decreased current flow. The composite electrode showed a sensitivity 4.8-times higher than that of the bare gold electrode. The biosensor was challenged with organophosphate-spiked food samples and showed a limit of detection (LOD) of DMT at 4.1 nM, with a limit of quantification (LOQ) at 12.6 nM, in the linear range of 10 nM to 1000 nM. Such performance infers significant potential for the use of this system in the detection of organophosphates in real samples.

## 1. Introduction

Organophosphorus compounds (OPs), including pesticides and chemical nerve agents, are harmful to human health. In agriculture, the uncontrolled and excessive use of OP can contaminate harvest crops and processed foods, and organophosphates are indeed widely used in crops for both human consumption and as animal feed. OPs, neuronal acetylcholinesterase (AChE) inhibitors, have been found to contaminate raw milk and infant formulas [1,2]. The presence of OP residues in food and dairy products has been reported to cause dopaminergic neurodegeneration [3,4], Alzheimer’s disease [5], Parkinson’s disease [6], and Amyotrophic Lateral Sclerosis (ALS) diseases [4]. The occurrence of neurodegeneration caused by OPs is associated with acute and chronic effects. The acute effect is associated with AChE inhibition, which causes both muscarinic and nicotinic toxicity due to the excessive accumulation of acetylcholine at the neuromuscular junctions and synapses [7]. The chronic effects are due to OP-induced free radical generation linked with enhanced oxidative stresses that become the key mechanism of their neurotoxic alterations in the long-term effects [8]. Although inhibiting AChE activity introduces considerable effects for OPs, previous studies found that OPs induce molecular alterations of neuron-associated targets, such as hormones [9], neurotransmitters [10], neurotrophic factor [11], and oxidative stress and mitochondrial dysfunction [12] in the chronic effect. These are increased occurrences of OP-induced developmental neurotoxicity and the age composition of the patient population [13,14]. Thus, the development of analytical methods to detect and quantify OP residues in food materials is very important in controlling food quality and preventing consequent health complications.

Neuronal AChE is a principal enzyme in the neurotransmitter pathway that hydrolyzes released acetylcholine in the brain in the periphery. The turnover number of AChE of 1.5 × 10^4^ s^−1^ makes it one of the most efficient oxidative/redox enzymes, allowing the catalysis of released choline in a submillisecond time frame [15]. The rapid catalysis of released acetylcholine is crucial to maintain the dynamic steady state between the synthesis and release of acetylcholine, which plays important roles in maintaining brain energy metabolism [16,17] and an energetic brain [18]. However, OPs inhibit AChE activity to cause an accumulation of acetylcholine in brains. The accumulated acetylcholine, which allows a higher occupancy rate and longer duration at its receptors, stimulates synaptic receptors and is involved in impaired acetylcholine-mediated neurotransmission [19].

Acetylthiocholine can be used as a substrate for AChE to produce thiocholine. Subsequently, the resulting thiocholine is electrolyzed to produce dithiobischoline and release two protons and two free electrons. This can be seen in the chemical reactions (i) and (ii) below. The AChE activity can be inhibited by OP; thus, OP levels can be determined through measurement of the reduced AChE activity of bioconverting acetylthiocholine (ATC) to thiocholine.

In electrochemical assays, the enzymatic thiocholine is electrolyzed (shown in (ii)) on electrode surfaces and less enzymatic thiocholine-less current is a basic platform to develop AChE-based OP detection biosensors. The platform can be run on electrochemical cells or microfluidic systems with monoenzymatic AChE or with a bienzymatic, choline oxidase-coupled AChE system [20,21].

Acetylthiocholine + AChE + H_2_O   →   Thiocholine + Acetic acid

(i)

Thiocholine + Electrolysis   →   Dithiobischoline + 2H^+^ + 2e^−^

(ii)


Many techniques have been used to detect and quantify various OPs in many kinds of samples, ranging from high-performance liquid chromatography (HPLC) [22] to mass spectroscopy [23,24], as well as immunoassays and chromogenic assays [25]. Although these techniques are considered standard analytical tools, their extended sample preparation delays readings and analysis must be performed in a laboratory setting with trained personnel, limiting their use. Alternatively, a potential option is to use AChE, one of the fastest enzymes [26], in developing electrochemical biosensing systems. Recently, AChE has been used to develop high-throughput screening platforms to explore novel drugs for neurodegeneration and neuromotor dysfunction [27,28,29]. AChE has also been used as a catalytic bioreceptor to develop electrochemical biosensors [30,31], providing broad applications in the detection of organophosphates in terms of the electrochemical change of electrode interfaces. In this work, we developed an AChE-based biosensor featuring carbon nanotubes (CNTs) on a poly(diallymethylammonium) (PDDA)-derived nanocellulose composite. CNTs in the composite structure serve to enhance electron transfer from free electron products of enzyme oxidative reactions to the electrode surface [32,33]. Therefore, the composite material strongly contributes to amperometric sensitivity. This material proved to have excellent immobilization of AChE as compared to bovine serum albumin (BSA)-coated gold electrodes. The results showed that AChE was mainly immobilized on the derived oxidized nanocellulose film via carbodiimide crosslinker chemistry (EDC/NHS activation). We compared the performance of the two electrodes on the fluidic chip to investigate enzyme inhibition patterns. Finally, the biosensor was tested for the detection of DMT in spiking samples to demonstrate the monitoring of DMT residues in food samples.

## 2. Materials and methods

### 2.1. Materials

Dimethoate (45449), poly(diallyldimethylammonium chloride) polymer (PDDA, 409022), EDC (1-ethyl-3-(3-dimethylaminopropyl)carbodiimide hydrochloride, 03450), N-hydroxysulfosuccinimide (NHS, 130672), acetylcholinesterase (1000 units/mg, C3389-500UN), acetylthiocholine (ATC, A5626), and 90% carbon basis multiwall carbon nanotubes (659258) were all purchased from Sigma-Aldrich (St. Louis, MO, USA). Thiocholine (ACM 645003) was purchased from Alfa Chemistry (Ronkonkoma, NY, USA). A 1 mg/mL stock solution of dimethoate and 100 mM ATC was prepared in DMSO and 0.01 M phosphate-buffered saline (PBS), respectively. The solutions were kept at −20 °C in aliquots for future use. Phosphate buffer (10 mM, pH 8.0) mixed with 0.1 M KCl was used, with KCl functioning as an additional supporting electrolyte. Other reagents were of analytical reagent grade, and solutions were prepared with ultrapure water having a resistivity of 18.2 MΩ∙cm.

### 2.2. Concentration-Response and Time-Response Study for AChE Inhibition

AChE-immobilized electrodes were exposed to different concentrations of DMT in the range of 0.001 to 5.0 μM. The concentrations selected were based on the results of DMT inhibition activities previously reported [34]. The inhibition pattern of DMT was measured under endpoint and real-time analysis. In endpoint analysis, we added 100 μL of 10 μM acetylthiocholine (ATC) to 10 mM phosphate buffer (pH 8.0) solution, as well as an additional supporting 0.1 M KCl electrolyte, after 2 min of exposure. Enzyme activity was determined by measuring thiocholine electrolysis-produced current.

### 2.3. Immobilization of AChE onto Composited Electrodes

Gold electrodes (1 × 1 mm^2^) were fabricated by the e-beam evaporation of 20 nm Cr and 200 nm gold (Au) onto SiO_2_/Si substrates, similar to those used in [35]. These electrodes were carefully washed in acetone by sonication for 5 min, rinsed with DI water, and dried under nitrogen gas. A 100 mg amount of oxidized nanocellulose (NC), prepared by the TEMPO-oxidization method [36], was used to disperse 0.1 mg of multiwalled carbon nanotubes (MWCNT) in isopropanol, forming a homogenous mixture of CNT-MWCNT. In this material, MWCNT was homogeneously distributed in a cellulose nanofibril network in order to achieve conductivity [37]. Subsequently, 20 µL of PDDA (20% in water) was added to disperse the CNT-NC composites, followed immediately by ultrasonic agitation to obtain a CNT-PDDA-NC mixture. The composite electrode was prepared by casting 5 µL of the CNT-PDDA-NC suspension onto the surface of the bare gold electrode and placing the electrode in a vacuum desiccator to evaporate the solvent. As controls, bare gold electrodes were treated with 0.5 mM 11-mercaptoundecanoic acid (11-MUA) and coated with 10 μL of AChE/BSA (ratio 1:1 *v*/*v*). A 5 µL volume of 1 mg/mL free AChE (pI 4.5–5.2, 71.6 kDa) in Tris–HCl buffer solution (pH 8.0) was immobilized via EDC/NHS bioconjugation [38] between carboxyl groups of oxidized nanocellulose on composite electrodes and 11-MUA linkers on gold electrodes.

### 2.4. Characterization of Modified Electrodes

Samples for scanning electron microscope (SEM) imaging were prepared by mounting the modified electrodes on carbon tape and sputter-coating them with approximately 5 nm of iridium (ACE600, Leica Microsystems, Buffalo Grove, IL, USA). Imaging was performed from 5 to 20 keV (GAIA3, Tescan, Czech Republic). Fourier transform infrared (FTIR) spectroscopic analysis was collected at a 1 cm^−1^ resolution over the wavenumber range of 4000–400 cm^−1^ versus a baseline correction (FTIR 4700, Jasco Inc., Easton, MD, USA). FTIR samples were prepared on microscope cover slides. Briefly, 30 µL of the composited substrate was dropped onto the surface of the cover slide and allowed to dry in a vacuum desiccator. X-ray photoelectron spectroscopy (XPS) was performed using a Kratos AXIS Supra photoelectron spectrometer (Kratos Analytical, Manchester, UK). Al Kα radiation was used to observe peak intensities for binding energy scans of C^1s^, N^1s^, and S^1s^. Adsorbed AChE on both the composite surfaces and bare gold electrodes were prepared to evaluate the biosensing performance.

### 2.5. Fabrication of Chip-Based Biosensor

The fluidic chip was composed of a six-layer sandwich structure made from laboratory Parafilm (PM-996), with patterned Parafilm layers placed between microscope slides to form an enclosed channel. The channel was prepared by thermally bonding the Parafilm and glass slide. The bonding process was performed on a glass slide at 45 °C using a hotplate with a manual press during the bonding process [39]. The bonded layer formed from heated Parafilm was cut directly to form the approximately 200 μm desired channels, as thin layers can be easily removed from the glass surface. The process is rapid, cost-effective, and does not require the use of complex procedures to fabricate simple fluidic systems [40]. A basic diagram of the channel structure and preparation steps are shown in Figure 1a,c.

The fluidic channel was manually templated and cut from the layer structure to make the fluidic channel. The working electrode (WE), reference electrode (RE), counter electrode (CE), and needles were placed at both ends to connect the device to a micropump. While the reference electrode maintains a constant potential, the working electrode measures the current during the potential scans. The counter electrode directs electricity from the signal source to the working electrode. The second microscope slide was pressed against the Parafilm layers, producing the said sandwich structure. The structure was evenly compressed while the chip was placed in an oven at 60 °C for 10 min. The chip was subsequently taken out and cooled to room temperature (25 °C). The four sides of the sandwich structure were then sealed by silicone sealant (Grainger, 53DA95) (Figure 1c).

### 2.6. Electrochemical Detection of Dimethoate

A desktop electrochemical analyzer (CHI 760E, CH Instruments, Austin, TX, USA) was used to perform the amperometric assay. Composite electrodes (1 mm^2^ surface area), Ag/AgCl (1 M NaCl) fritted glassy electrodes, and exposed platinum wire electrodes served as the WE, RE, and CE, respectively. A syringe pump (Chemyx Fusion 200, Provac, Stafford, TX, USA) delivered the phosphate buffer supporting electrolyte at a flow rate of 100 μL/min, in order to achieve a steady-state current prior to injection of ATC solution.

In cyclic voltammetric (CV) assays, the potential was cycled between −0.6 and 0.8 V, with a scan rate of 0.2 V/s and a sample interval of 0.001 V, for both the composited electrode and bare Au electrode. The limit of detection and inhibition patterns were determined from concentration-based assays for the substrate (ATC) and the inhibitor (DMT), respectively. In amperometric measurements, the working potential was set at 150 mV, and transient currents as a function of time were measured. Sensitivity calibration curves for 0.01–50 μM of ATC were obtained by successive dilution of the electrolyte-buffered PBS (10 mM, pH 8.0) [41]. Linear regressions (μA vs. [ATC] and μA vs. [DMT]) were performed to determine LOD values and inhibition patterns of DMT on AChE. The limit of detection (LOD) was determined according to IUPAC recommendations [42].

After achieving a steady-state current, 50 µL of 10 μM ATC was injected into the device to collect initial AChE activity in the absence of DMT. Subsequently, various concentrations of DMT ranging from 0.01 to 1.0 μM DMT were injected. When the solution reached the active cavity span of the biosensor, the flow was stopped. The fluidic chamber was then closed and the solution containing DMT was incubated for 2 min for DMT to bind to immobilized AChE active sites. Following the inhibition step, the response of the composited electrodes to electrons from thiocholine electrolysis was recorded.

Thiocholine is the enzymatic product of AChE and ATC, where less is produced in the case of DMT-inhibited AChE. This leads to a decrease in the number of electrons created. Based on a linear relationship between the amount of AChE-produced thiocholine and the electrical current, the amount of DMT in the sample was calculated. As noted, adding ATC (or DMT) to the fluidic chamber either produces (or inhibits) enzymatic thiocholine production. The resulting current was recorded over time, enabling a “fed-batch” process with a reaction time of 2 min (including the loading time). Because K_cat_ is equal to a turnover number, defined as the number of molecules of thiocholine made per AChE per second, the “fed-batch” process allowed for the determination of the K_cat_ values of AChE activity: 1.5 × 10^4^ ATC molecules per second for each AChE active site [43].

### 2.7. Preparation of Food Samples

To test a variety of real-world scenarios, DMT-spiked orange juice, coffee, and 2% fat milk were chosen as evaluation candidates. DMT concentrations of 0.01 μM and 0.1 μM of DMT were mixed into 1 mL of each sample. The samples were injected into the biosensor chip and kept at room temperature for 2 min, then 10 μM of ATC prepared in 10 mM PBS (pH 8.0) and 0.1 M KCl was injected without being treated with DMT. The presence of DMT in the spiked samples was detected by observing the current decrease.

### 2.8. Analysis of Inhibition Patterns of Dimethoate against AChE

Inhibition patterns were determined by measuring the remaining activity of AChE after a rapid and large dilution of the AChE-DMT complex. Inhibition patterns of DMT were calculated by how the AChE-DMT complex influences the steady-state reaction velocity equation [44]:(1)$$\upsilon =\frac{V_{max}\left[ACT\right]}{\left[S\right]+\;K_{M}\left(1+\frac{\left[DMT\right]}{K_{i}}\right)}$$
where the $K_{M}\left(1+\frac{\left[DMT\right]}{K_{i}}\right)$ term = K_M,apparent_ (determined by experimental assay), and K_i_ is the actual enzyme–inhibitor I complex dissociation constant.

The inhibition percentage of DMT to AChE activity is defined as:(2)$$\%\;inhibition=100\;\times \;\left(1-\frac{v_{i}-I_{b}}{v_{o}-I_{b}}\right)$$
where and $v_{o}$ are the reaction velocity in the presence and absence of DMT at 3 ppb (~13.1 nM), respectively, and $I_{b}$ is the current background change with time.

Based on the fractional activity:(3)$$\frac{v_{i}}{v_{o}}=\frac{1}{1+\left(\left[DMT\right]/IC_{50}\right)}$$
and using mass–balance relationships between AChE and DMT (or $1-\left(\frac{v_{i}}{v_{o}}\right)$), % inhibition of DMT becomes:(4)$$\%\;inhibition=\frac{100}{1+\left(\;\frac{IC_{50}}{\left[I\right]}\;\right)}\;$$

The differences in enzyme activity measured in resulting current with or without the presence of an inhibitor form the basis of analyte detection, according to:(5)$$\%\;inhibition=100\;\times \;\frac{I_{0}-I_{i}}{I_{0}}$$
where $I_{0}$ and $I_{i}$ are the current in the absence and presence of DMT, respectively.

## 3. Results and Discussion

### 3.1. Characterizations of AChE-Modified Electrode

Enzyme immobilization patterning in self-assembled molecular monolayers (SAMs) has long been reported as a simple and powerful method to construct layered redox enzymes and solid electrode surfaces [45]. AChE molecules immobilized via EDC/NHS bioconjugation, as well as their electrostatic interactions with electrode surfaces, are shown in Figure 1b. Previous studies revealed that the charged PDDA substrate also strongly contributes to the adsorption of AChE through electrostatic interactions [46,47]. These enzyme layers, in turn, allow DMT molecules to access the active sites of AChE, which inhibit AChE’s activity. The electrostatic adsorption maintains the native structure of the AChE molecules and allows their active sites to contact targets (substrates or inhibitors) [48], with the enzymatic reaction then occurring on the surface of the electrodes.

An SEM image of the AChE-immobilized gold electrode surface is shown in Figure 2a and AChE-coated composited electrodes in Figure 2b. AChE operating as the bioreceptor was immobilized by both covalent and ionic adsorptions. The AChE immobilized onto Au occurred through a well-known bio conjugative link, carbodiimide, between 11-MUA molecules and BSA/AChE complexes. BSA was used to protect AChE activity by creating a 3D network for enzyme entrapment [49]. For the composited electrode, the functionalized matrix structure of the composited electrode can be observed in the SEM image (Figure 2b).

Enzyme and organic elements on the electrode surface were characterized by FTIR (Figure 2c). The spectra shown are transmission spectra of the electrode catalyst layers affixed to round microscope slides, and there are multiple peaks of interest. Strong carbonyl stretches of amide I and II (in 1740 and 1702 cm^−1^ peaks of the spectra), resulting from secondary amides of the crosslinkers of immobilized AChE, are consistent with the C-N-H stretch bend of a monosubstituted amide [50]. Peaks of amide III at 1305 and 1244 cm^−1^ (C-N) demonstrate the α-helix and β-sheet in the protein structures of BSA and AChE [51]. An additional strong peak at 1672 cm^−1^ is due to an in-plane N-H bend of the primary amide. The weak peak at 1303 cm^−1^ is due to a carbonyl stretch of oxidized nanocellulose, as well as the conjugated crosslinkers produced by the bioconjugation. The broad peak at 3227 cm^−1^ and shoulder at 3206 cm^−1^ are due to N-H antisymmetric and symmetric stretching, respectively. The doublet at 2933 and 2922 m^−1^ is due to antisymmetric and symmetric CH_2_ stretches, respectively, present in 11-MUA and nanocellulose. A peak at 2832 cm^−1^ is also due to the symmetric bending of a coupled thiol group. The strong adsorption bands of amide I and amide II (3227 cm^−1^ N-H stretch), as well as NH and NH_2_ bands, are characteristic of the enzyme, indicating the AChE was successfully coated on the composited and Au surface. Biosensing surface analysis was performed by XPS (Figure 3). Surface compositions were obtained by scans of the Au substrate electrode, including a survey scan (Figure 3a) and targeted scans of the S^2p^ (Figure 3b), C^1s^ (Figure 3c), and N^1s^ (Figure 3d) orbitals. The survey scan demonstrated an Au^4f^_7/2_ peak at 84.1 eV, indicating the presence of bonded Au. The scan of a control sample showed an Au^4d^_3/2_ peak at 340.7 eV, indicating a bare Au surface. The S^2p^ peak was observed to have a spin-orbit splitting doublet for S^2p^_1/2_ and S^2p^_3/2_. The first peak, centered at about 160.5 eV, is assigned to the sulfur moieties of 11-MUA on the electrode surface, which is related to the chemisorption of thiols. The second peak at about 162 eV is associated with the physisorption of free sulfur moieties on the surface. The C^1s^ spectra, resulting mainly from the carbon nanotubes (CNT), featured several indicative peaks: the nonoxygenated ring C1 (283.9 eV) that covers the C = C bond of hybridized sp2, the C in C-O bond at 285.4 eV that results from hydroxyl groups in nanocellulose, and the C in C-O bonds at 286.4 eV that result from carboxyl groups [52].

The result clearly shows a transition of binding energy from -C-O-H (hydroxyl groups) to -C = O (in carboxylic acid groups) in oxidized MWCNTs, which reflects oxidation degree. In addition, the spectra of N^1s^ represent carbodiimide bonds that formed to crosslink carboxylic acids to side chain lysine in AChE molecules during EDC and NHS conjugations. The N^1s^ spectra show two distinct peaks at 398.7 and 399.9 eV, allocated to the N-atom of carbamide and amide ([N-C]O) [53], respectively. The attribution is mostly influenced by the binding of AChE to the electrode surface, suggesting a shift from free amine at 399.3 eV to carbamide bonds at 398.7 eV. The curve fitting of the N^1s^ narrow scan revealed amide groups at 405.9 eV, suggesting the presence of unreacted NH_2_ side chain groups of AChE.

### 3.2. Measuring AChE Activity on Electrodes

Through amperometry measurements, catalytic activity and inhibition patterns (reversibility and irreversibility) of ATC and DMT were measured, respectively. DMT inhibits AChE catalytic activity, where the amount of enzymatic thiocholine product is reduced during the inhibition process. On both bare Au and composited electrodes, resulting enzymatic thiocholine is well known to adsorb through Au-S covalent bonding. This, in turn, promotes the electrolysis process of enzymatic thiocholine on electrode surfaces [54,55]. Cyclic voltammetric (CV) measurements were conducted by cycling the potential between −0.6 and +0.8 V (versus Ag/AgCl 0.1 KCl) for three consecutive scans at a scan rate (υ) of 0.2 mV/s. Results for the AChE-coated composite electrodes are shown in Figure 4a and results for the AChE-coated Au electrode are shown in Figure 4b.

Because AChE specifically catalyzes ATC substrates and produces enzymatic thiocholine at the electrode interface, the larger active surface area of the composite electrode over the bare Au electrode is expected to increase the overall AChE catalytic activity. CV scans of the composite electrode show comparable reductive current, along with significant current sensitivity to the concentration of thiocholine, as compared to AChE-modified Au electrodes. A pair of rather well-defined redox peaks, E_pc_ + 0.24 V and E_pa_ +0.31, appeared regularly in the scans. The reversible reduction peak corresponds to the reduction of the thiol group to form dithiobischoline. This reaction pathway is consistent with those described elsewhere for acetylthiocholine bioconversion [56]. Commercial thiocholine was used in the control experiment under the same sensing conditions with enzymatic thiocholine to determine CV peaks. Expected redox peaks appeared within the selected potential range of −0.6 to +0.8 V (Appendix A). The current response of the composited electrode is 4.5 times higher than that of the Au electrode, as calculated from the intensity of E_pa_ and E_pc_ peaks, up to a larger active surface area. Moreover, the significant enhancement in the E_pa_ peak of the thiocholine may be due to fast electron transfer through the composite layers. In AChE-modified electrodes, a lower current response was observed (Figure 4b), assumed to be a result of slow electron transfer through BSA layers due to a low content of amino-acid-containing phenol backbones (e.g., tyrosine) [57,58]. Previous studies [32,59,60,61] have demonstrated that multiwall CNTs efficiently promote the electrocatalytic oxidation of thiocholine. As shown in chemical pathways (i) and (ii), thiocholine products undergo rapid dimerization to form the dithiobischoline dimer species. pH was set at 7.4 to eliminate the pH effect on electrocatalytic oxidation of thiocholine. Previous studies have demonstrated that a more basic pH promotes thiocholine oxidation due to deprotonation [62].

The effect of scan rate on the electrochemical oxidation of thiocholine at surface electrodes has been reported previously [63,64] and occurred similarly here. Relationships between the oxidation peak current (I_p_) and reduction peak current (I_a_) are shown in Figure 4c. Both the oxidation and reduction peaks increased gradually with the sweep rate, shifting to more positive and negative values, respectively. A linear fitting line was obtained by plotting the current peaks as a function of the square root of scan rates, demonstrating that electrolysis was a diffusion-controlled process [65,66].

A potential of 150 mV was selected for amperometric measurements [67], conducted in the same conditions as cyclic voltammetric measurements. The amperometric response of thiocholine on the composited electrode is shown in Figure 4d. ATC concentrations ranged from 0.1 to 50 μM, as there was no discernible current response at concentrations less than 0.1 μM. Concentration-based results show that the immobilized AChE exhibited rapid and specific catalysis to convert ATC to thiocholine (Appendix A). With composite electrodes, the biosensor obtained 93% of the steady-state current within 76 s, which is slower than reported values closer to 15 s [67]. This gap may be due to less diffusion of enzymatic thiocholine or lower electron-transfer rates on the electrode surface. In this platform, an insufficient loading of AChE affects biosensor performance, as it is dependent on the rate of the enzymatic bioconversion from ATC to thiocholine. With BSA-coated electrodes, lower electron-transfer rates on its surface due to the BSA layer [44] affect the current response. These considerations can be overcome by introducing enzyme loading [68] of the native conformational AChE and enhancing electron-transfer rates on the sensor surface [32] during electrocatalysts.

### 3.3. Detecting Dimethoate and Determining Inhibition Patterns

Figure 5a shows the decreased current when DMT concentrations were increased. The reversible oxidation peak was found at +0.31 V, associated with the oxidation process of thiocholine. As expected, increases in DMT reduce the current. One important note when analyzing Figure 5a is that although the oxidation peak is not visible when DMT is at 1.0 µM, the current dropped from 6.3 to 5.5 μA when 1.0 μM DMT was introduced due to the increased inactivity of AChE (Figure 5b and Appendix A). The amperometric response of the AChE-based biosensor for DTM detection originates from the oxidation current of thiocholine. This performance of the biosensor should significantly rely on the inhibition of DMT against AChE activity. Figure 5b showed the typical current versus time during different DMT concentrations (1.0–1000 nM) added in the sensor. The result shows a significant decrease in AChE activity through observing dropped current peaks.

With a continuous assay, by adding a fixed amount of ATC (10 μM) at 120 s intervals, the current plateaued after 1750 s at 14.3 μA (Figure 5c, left). This behavior is related to the high affinity of DMT to AChE active sites. Rates of AChE inactivation were measured by adding a constant flow of 10 μM and 0.1 μM solutions of ATC and DMT, respectively. The decreasing current intensity may be due to the dissolution of AChE from the electrode surface. At saturation, the aforementioned ATC/DMT mix was injected, and the flow was stopped for 120 s. The residual activity of AChE was influenced by its exposure time to DMT, as its active sites should still have been occupied by DMT. Likely for this reason, the amperometric response quickly decreased from 2000 to 3000 s after exposure (Figure 5c, right).

Binding of DMT to AChE was observed from the dependence of the pseudo-first-order association rate, kobs of the reversible inhibition of DMT on AChE active sites [69,70] on DMT concentrations. A plot of the kobs (obs means “observed rate constant”) versus [DMT] allowed for the determination of the Ki value for a time dependent DMT. Figure 5d shows the relationship between Kobs and [DMT], shown in (6)
(6)$$k_{obs}=\;k_{3}\left[DMT\right]+\;k_{4}$$
where $k_{3}$ and $k_{4}$ are determined from the slopes and y-intercepts of Figure 5d, respectively. The slope of each plot was the k_obs_. K_obs_ values, indicating that the interaction between organophosphate with AChE is a slow binding inhibition [69].

As a result, a slow establishment of AChE-DMT equilibrium occurs with a slow onset of inhibition prior to reaching the steady state [71], which provides the pattern of DMT inhibition and kinetic constants for binding and association of DMT.

The velocity of the AChE reaction (Reactions (i) and (ii)) is proportionally responsible for the activity rate of the immobilized AChE on the electrode surface. The plot in Figure 6a presents decreased currents versus DMT concentrations obtained during amperometric detection with the composited electrode. Because significant background noise was present during measurement, nonlinear regression was used to fit the enzyme catalysis data [72]. The current decreases rapidly with the increase of DMT concentration. When the DMT concentration reaches 0.47 μM or higher, the current response tends to reduce more rapidly, which can be explained by the maximum occupation of DMT in AChE active sites. The LOD values (inset of Figure 6a) were calculated to be 4.1 ± 0.16 nM and 16.9 ± 0.06 nM (in PBS buffer) for the composited electrode (E1) an BSA-coated Au electrode (E2), respectively (Figure 6a, top right). The formation of conductive nanostructures on the surface of the composite electrode provides a favorable microenvironment for AChE activity and increases the electrolysis rate for thiocholine compared to the BSA-coated Au electrode. Although the LOD value of this work is higher than those of previous reports (Table 1), the biosensor can work well on those fresh vegetable samples, which are contaminated by higher 39.2 nM of organophosphate residues (here it is for dimethoate) [73]. Moreover, fast, simple, and cost-effective fabrication could be considered a competitive advantage for mass production.

The detection of DMT in spiked samples of milk, coffee, and orange juice is shown in Figure 6a. The standard deviation (SD) of the current measured by the BSA-coated Au electrode is much lower (denoted as E2) (5.6%) compared with that of the composited electrode (denoted as E1) (15.6%). The difference indicates that the BSA-coated Au electrode can effectively prevent the dissolution of AChE and provide good reproducibility and stability. Surface chemistry and bioconjugation processes in the fabrication of composite electrodes will be further considered in future work.

The titration of ATC to determine the steady-state velocity for AChE was performed; DMT concentrations from 0 to 5 μM were used. The K_M_ constant (Michaelis–Menten) was calculated by using the Lineweaver–Burk plot of 1/v versus 1/[ATC] (Figure 6b). A value of K_M_ at 0.4 (μM) was determined at 0 μM of DMT. As shown in Figure 6b, the (1/V_max_) value is constant at 2.5 for all DMT concentrations, but the apparent value of K_M_ (K_M_/V_max_) increases with increasing DMT concentration. This result demonstrates that DMT inhibits AChE under competitive inhibition. A previous study reported DMT molecules bind to the active site (serine 200, histidine 440, and glutamate 327) of AChE by forming a covalent bond with the serine residue at the active site [81]. DMT molecules bind to other sites of an AChE molecule (Appendix A) and its active site (Appendix A) via molecular Swiss docking methods [82]. However, DMT and ATC could bind to separate sites on AChE that somehow exert a negative regulation on one another through negative allosteric interaction, which could be explained in further investigations.

## 4. Conclusions

We present a sensitive fluidics-based amperometric biosensor for detecting DMT, a typical OP, working on composited electrodes. The formation of conductive nanostructures on nanocellulose matrices provides a promising microenvironment for retaining AChE activity. The electrocatalytic promotion of MWCNT contributes to enzymatic thiocholine electrolysis on the electrode surface. The composited electrode integrated into a fluidic chip was used to detect DMT as low as 4.1 nM DMT. The developed biosensor shows good sensitivity, precision, and reproducibility (but less than BSA-coated Au electrodes, as compared) for DMT-spiking samples. However, this biosensor design features significant advantages in the simplicity of the fabrication and provision of an effective immobilization platform for enzymes. This platform could be used to study enzyme assays of inhibitors for a wide range of applications.

## Figures and Tables

**Figure 1 micromachines-12-00397-f001:**
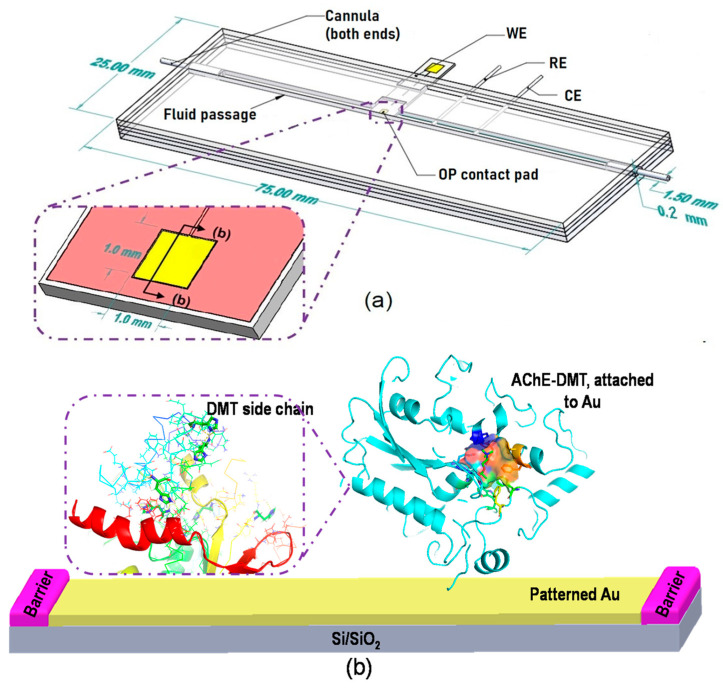
Schematic illustration for the biosensor development. (**a**) Three-electrode configuration is integrated into a fluidic chip for dimethoate (DMT) detection. Inset box is the working space (WE) where acetylcholinesterase (AchE) is immobilized. (**b**) Immobilization of AChE on working electrodes. AChE shows active sites and DMT molecules occupy the active sites to inhibit AChE activity. (**c**) Schematic of preparation steps for making channels for flow control.

**Figure 2 micromachines-12-00397-f002:**
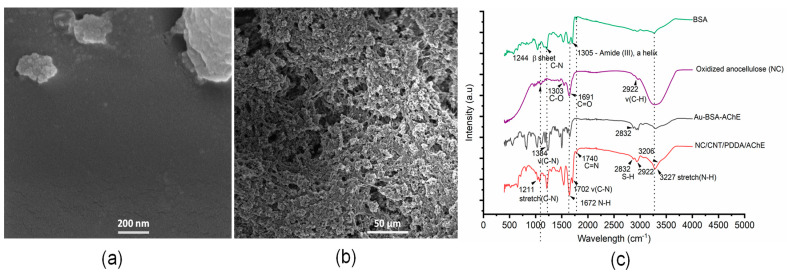
Surface characterization of the electrodes. Scanning electron microscope images of the (**a**) AchE/BSA-coated gold electrode and (**b**) AchE/MWCNT/PDDA/NC on the surface of gold electrodes. (**c**) FTIR spectra of organic elements on the electrode surface.

**Figure 3 micromachines-12-00397-f003:**
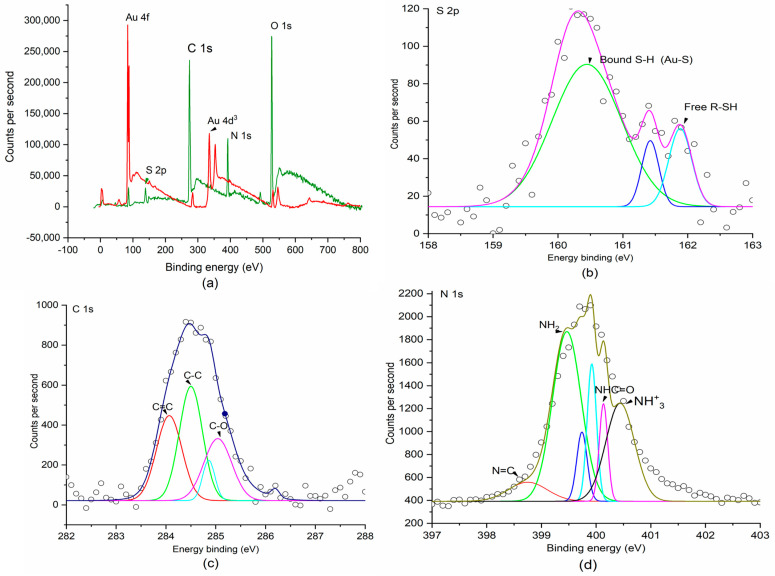
Binding energy evaluation of elements on the electrode surface. (**a**) X-ray photoelectron Scheme 2, (**b**) S 2p, (**c**) C 1s, and (**d**) N 1s. Al Kα radiation used to observe matching peak intensities for the binding energy scan.

**Figure 4 micromachines-12-00397-f004:**
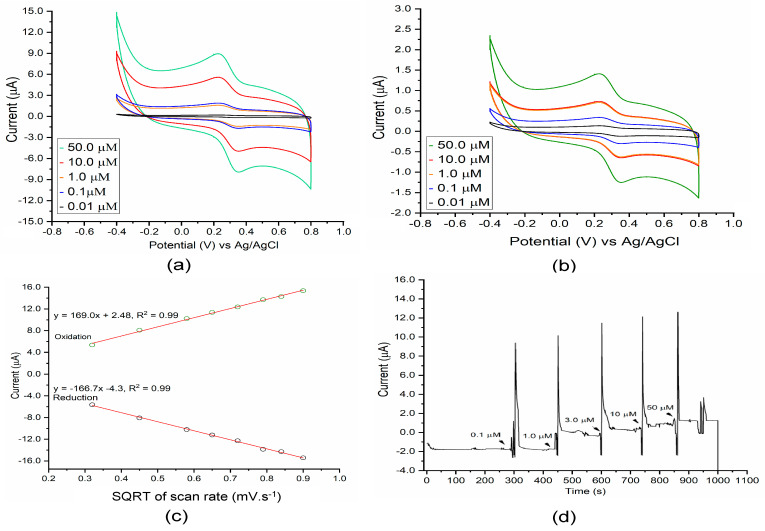
Cyclic voltammograms of (**a**) composited electrodes and (**b**) BSA-coated gold electrodes in 10 mM phosphate buffer supported with 0.1 M KCl. Scanning rate (υ): 0.2 mV/s. (**c**) Corresponding plots of cathodic and anodic peak currents versus the square root of potential scan rate from (**a**). (**d**) Amperometric responses of the composited electrode with successive addition of ATC solution under fluidic conditions in PBS pH 8.0 supported with 0.1 M KCl. Potential of 1.5 V vs. Ag/AgCl was applied on the composited sensor. The working solution was the phosphate buffer (10 mM, pH 8.0) mixed with 0.1 M KCl.

**Figure 5 micromachines-12-00397-f005:**
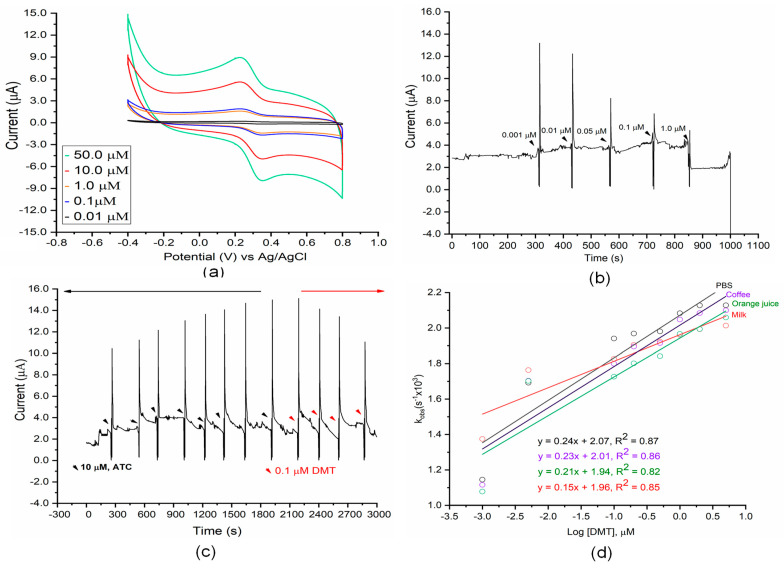
Electrochemical assay of DMT-inhibited AChE assays. (**a**) Cyclic voltammograms of the fluidic biosensor in presence of different concentrations (0.01–1.0 μM) of DMT. (**b**) Amperometric responses of the composited electrode for DMT analysis. (Flow rate: 100 μL/min; working potential: 150 mV). (**c**) shows the decreased activity of AchE due to the fixed substrate ATC (10 μM) (left) and accumulated DMT (0.1 μM) in the fluidic biosensor (right). Amperometric assay with a potential of 150 mV on the composited sensor. The assay was run in the phosphate buffer (10 mM, pH 8.0) mixed with 0.1 M KCl. (**d**) pseudo-first-order rate constant as a function of DMT concentration in different spiked samples (Equation (6)).

**Figure 6 micromachines-12-00397-f006:**
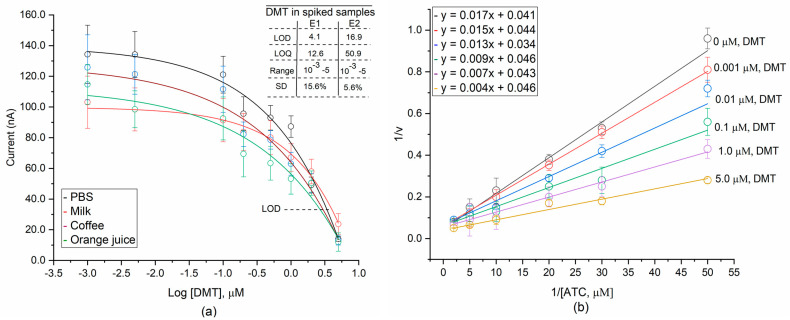
Determination of DMT inhibition patterns. ATC titration of steady-state velocity of AChE in the presence of different concentrations of DMT in PBS, milk, coffee, and orange juice. (**a**) A correlation between generated currents versus DMT concentration (in logarithmic base 10). (**b**) Lineweaver–Burk plot for determination of the inhibition pattern of DMT to AChE in the spiking solution of PBS E1: composited electrode, E2: BSA-coated electrodes.

**Table 1 micromachines-12-00397-t001:** Comparative Performances of different biosensors in organophosphate detections.

Sensing Receptors	Sensing Methods	Detection Limit
Aptamers, enzymes	Localized surface plasmon resonance (LSPR)	58–1835 ppm [74]
	Fluorescence	1 mg mL^−1^ [75]
	Nanoplasmonic	10 ppb [76]
	Cyclic voltammograms	0.2 nM [52] 8.0 pM [77]
	Amperometry	0.27 μM [78,79] 9 nM [56]
	Field-effect transistor	1.8 fM [80]

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
