# Peer review of "A Fluidics-Based Biosensor to Detect and Characterize Inhibition Patterns of Organophosphate to Acetylcholinesterase in Food Materials"

_micromachines, 2021, doi:10.3390/mi12040397_

Round 1
Reviewer 1 Report
Dear Editor
The study by Pham and Nguyen et al describes a chip-based electrochemical biosensor for the detection of organophosphate in food materials using composited electrodes.
The design of the study and the technical quality of the work are convincing and results can be of general interest since it can be widely applied. The methods were mostly well described and it was good to see that the authors have successfully investigated both of the time- and concentration- dependent effects.
However, there is a number of major and minor points that would need to be addressed in order to improve the quality of this paper before it can be accepted for publication:
Major:
-There was no description about the flow in their system. How it’s controlled, the flow rate and what are the measures that the authors have implemented to overcome some of the common issues about microfluidic systems including air-bubbles. Moreover, the authors have mention in p4 line 16 that they have used “silicone sealant” but they didn’t specify what type nor mentioned how they overcome some challenges like surface adsorption on similar materials like PDMS (which needs organic extraction). Technical details like these need to be mentioned and discussed in the light of similar work: https://pubmed.ncbi.nlm.nih.gov/33117784/
-The introduction doesn’t provide enough background and the novelty of the work wasn’t well-articulated.
- P1 line 36-37 “The presence of OP residues in food and dairy products has been reported to cause various neurodegenerative diseases”: which ones? It’s better to briefly define them and mention that they result from neuronal death. References to be included:
https://pubmed.ncbi.nlm.nih.gov/26687930/
https://pubmed.ncbi.nlm.nih.gov/27262357/
- P2 line 1-3 “where its catalytic and inhibition activity are exploited to develop electrochemical biosensors”. Authors need to firstly introduce different types of assays including high-throughput screening before zooming in to why they have preferred one over another. References to be included:
https://pubmed.ncbi.nlm.nih.gov/16737790/
https://pubmed.ncbi.nlm.nih.gov/33672148/
- P2 line 6-8 “The turnover number of AChE of 1.5 x 104 s-1 makes it one the most 6 efficient oxidative/redox enzymes, allowing catalysis of released choline in a sub-millisecond time frame”. The authors have touch upon an important point and it will be good if they can expand on the potential role in the energetic brain and brain energy metabolism. References to be included:
https://pubmed.ncbi.nlm.nih.gov/31318452/
https://pubmed.ncbi.nlm.nih.gov/21382422/
- The last paragraph of the introduction isn’t well linked to the rest. Authors would be encouraged to use this part to identify the main current challenge and introduce how their new approach would help towards a solution.
Minor:
-Figure 1 is a bit crowded, specially the zoom in part (a). A clearer version will be needed.
-P10 line 39: typo- assay.
Best.
Author Response
Article
A Fluidics-based Biosensor to Detect and Characterize Inhibition Patterns of Organophosphate to Acetylcholinesterase in Food Materials
Dang Song Pham*1, Xuan Anh Nguyen*1, Paul Marsh2, Sung Sik Chu1, Michael P.H. Lau3, Anh H. Nguyen¶2,3, Hung Cao¶1,2,3
1 Biomedical Engineering Department, University of California Irvine, USA
2 Electrical Engineering and Computer Science Department, University of California Irvine, USA
3 Sensoriis, Inc. 7500 212th St SW Ste 208 Edmonds, WA, USA
* Equal contribution
¶ Co-correspondence
1 Affiliation1; [email protected]
2 Affiliation2; [email protected]
* Correspondence: Correspondence: [email protected]; Tel: +1 (949) 824-8478
Hung Cao
Anh H. Nguyen
Electrical Engineering and Computer Sciences
University of California, Irvine, CA 92617
Rebuttal letter
Dear Editors and Reviewers,
We are thankful for the generous comments on the manuscript and we have edited the manuscript to address the concerns.
The reviewers’ comments are in black, while the author's response is in blue, and revised phrases are in red in the revised manuscript.
Review report 1
The study by Pham and Nguyen et al describes a chip-based electrochemical biosensor for the detection of organophosphate in food materials using composited electrodes.
The design of the study and the technical quality of the work is convincing, and results can be of general interest since it can be widely applied. The methods were mostly well described, and it was good to see that the authors have successfully investigated both of the time- and concentration-dependent effects.
However, there is a number of major and minor points that would need to be addressed in order to improve the quality of this paper before it can be accepted for publication:
Comment 1. There was no description about the flow in their system. How it’s controlled, the flow rate and what are the measures that the authors have implemented to overcome some of the common issues about microfluidic systems including air-bubbles. Moreover, the authors have mention in p4 line 16 that they have used “silicone sealant” but they didn’t specify what type nor mentioned how they overcome some challenges like surface adsorption on similar materials like PDMS (which needs organic extraction). Technical details like these need to be mentioned and discussed in the light of similar work: https://pubmed.ncbi.nlm.nih.gov/33117784/
Answer. Thank you for your detailed comments. The flow rate and other hydrodynamic phenomena including (i) buffer solution slipping over the solid surface, (ii) air-bubbles formation in the channels, (iii) chaotic mixture of a reaction solution in the channel, (iv) multiphase flow in the channel, and (v) counter-intuitive effect in microfluidics (the bottle neck effect) are considered during design the chip.
(i) Two sides of glass slides were treated with 100 mM 3-aminopropyltriethoxysilane (APTES) to convert its naturally hydrophobic surface to a hydrophilic surface.
(ii) Since this is a single-channel fluidic system, the air-bubble formation was removed by keeping the flow of solution until the trapped air is expelled from the channel.
(iii) The solutions we used are ATC or DMT that was well mixed before separately injecting into the fluidic system. The well-mixed solution allows ATC and DMT molecules to have an equal chance to access the active sites of AChE. Thus, the chaotic mixture of the solution was not crucial in this case.
(iv) We could consider this phenomenon if the mixture of ATC and DMT was injected simultaneously through the fluidic system. To avoid forming the multiphase flow, we well mixed the mixture before injecting it into the one-channel fluidic system. Thus, multiphase flow could be eliminated in the system.
(v) The bottleneck effects are present at the inlet and outlet of the single fluidic channel, but this phenomenon does not affect to the catalytic activity of the enzyme on the electrode surface since the inlet and outlet were closed after the bubble removal.
Regarding the term “silicon sealant”, we used Grainger Clear Sealant, Silicone (Grainger 53DA95) to seal outside edges of the fluidic system. The materials that contacted ATC and DMT dissolved in solution were parafilm and APTES-treated glass slides. They were used to make the single-channel (please see the image below). The technical details of this simple fabrication are added and discussed in the revised manuscript. Please refer to the revised format on page 4 lines 16 to 25 for your consideration.
We also added a flow dynamic simulation at a flow rate of 100 µL/min in the fluidic system for your consideration.
Comment 2. The introduction doesn’t provide enough background and the novelty of the work wasn’t well-articulated.
Answer. Supporting information was added in the introduction section of the revised manuscript. Please refer to lines 34-38 and line 51 of page 1.
The presence of OP residues in food and dairy products has been reported to cause dopaminergic neurodegeneration [1], Alzheimer's disease [2], Parkinson's disease [3], and Amyotrophic Lateral Sclerosis (ALS) diseases [4]; thus, development of analytical methods to detect and quantify OP residues in food materials is very important in controlling food quality and prevent consequent health complications. Acetylcholinesterase (AChE) is a principal enzyme in the neurotransmitter pathway that hydrolyzes acetylcholine, which is one of the most important neurotransmitters in the nervous system. However, OPs inhibit AChE activity to cause an accumulation of acetylcholine in humans. The accumulated acetylcholine stimulates synaptic receptors and damages the neuron system [5].
Comment 3. P1 line 36-37 “The presence of OP residues in food and dairy products has been reported to cause various neurodegenerative diseases”: which ones? It’s better to briefly define them and mention that they result from neuronal death. References to be included:
https://pubmed.ncbi.nlm.nih.gov/26687930/
https://pubmed.ncbi.nlm.nih.gov/27262357/
Answer. The phrase “The presence of OP residues in food and dairy products has been reported to cause various neurodegenerative diseases” at lines 36-37 was added some detailed information. The new phrase is “The presence of OP residues in food and dairy products has been reported to cause dopaminergic neurodegeneration [1], Alzheimer's disease [2], Parkinson's disease [3], and Amyotrophic Lateral Sclerosis (ALS) diseases [4].” The references were also added. Please refer to lines 34-36 in the revised manuscript for your consideration.
Comment 4. P2 line 1-3 “where its catalytic and inhibition activity is exploited to develop electrochemical biosensors”. Authors need to firstly introduce different types of assays including high-throughput screening before zooming in to why they have preferred one over another. References to be included:
https://pubmed.ncbi.nlm.nih.gov/16737790/
https://pubmed.ncbi.nlm.nih.gov/33672148/
Answer. Lines 1-3 were revised, stated as “Alternatively, a properly designed biosensing system may be a potential option to address all the above-mentioned issues. Biosensors in this field are mainly associated with acetylcholinesterase (AChE) scenarios. First, AChE has been used to develop high-throughput screening platforms to discover novel drugs for neurodegenerative diseases [6, 7]. Second, AChE has been used as catalytic receptors to develop electrochemical biosensors [8, 9].
Comment 5. P2 line 6-8 “The turnover number of AChE of 1.5 x 104 s-1 makes it one the most 6 efficient oxidative/redox enzymes, allowing catalysis of released choline in a sub-millisecond time frame”. The authors have touch upon an important point, and it will be good if they can expand on the potential role in the energetic brain and brain energy metabolism. References to be included:
https://pubmed.ncbi.nlm.nih.gov/31318452/
https://pubmed.ncbi.nlm.nih.gov/21382422/
Answer. We expanded the important point on the potential role in the energetic brain as the reviewer suggested. Please refer to line 43-49 of page 1 for your consideration.
“The turnover number of AChE of 1.5 x 104 s-1 makes it one the most efficient oxidative/redox enzymes, allowing catalysis of released choline in a sub-millisecond time frame [10]. The rapid catalysis of released acetylcholine is crucial to maintain the dynamic steady state between synthesis and release of acetylcholine, which plays important roles in maintaining brain energy metabolism [11, 12] and the energetic brain [13]. However, OPs inhibit AChE activity to cause an accumulation of acetylcholine in brains. The accumulated acetylcholine, which allows a higher occupancy rate and longer duration at its receptors, stimulates synaptic receptors and involve in impaired acetylcholine-mediated neurotransmission [5].”
Comment 6: The last paragraph of the introduction isn’t well linked to the rest. Authors would be encouraged to use this part to identify the main current challenge and introduce how their new approach would help towards a solution.
Answer. Thank you for your suggestion. We edited the last paragraph to get well-linked to the rest. Current challenges and new approaches based on biosensors were added accordingly. We quote the last paragraph here for your consideration.
“Many techniques have been used to detect and quantify various OPs in many kinds of samples, ranging from high-performance liquid chromatography (HPLC) [14] to mass spectroscopy [15, 16] and even immunoassays and chromogenic assays [17]. Although those techniques are considered standard analytical tools, their extended sample preparation delays readings, and the analysis must be performed in a laboratory setting with trained personnel, limiting their use. Alternatively, a potential option is to use AChE, which is one of the fastest enzymes [18], in developing electrochemical biosensing systems. Recently, AChE has been used to develop high-throughput screening platforms to explore novel drugs for neurodegeneration, neuromotor dysfunction [6, 7, 19]. AChE has also been used as catalytic receptors to develop electrochemical biosensors [8, 9], which provide broad applications in the detection of organophosphates in terms of electrochemical change of electrode interfaces. In this work, we developed an AChE-based biosensor featuring carbon nanotubes (CNTs) on a poly(diallymethylammonium) (PDDA)-derived nanocellulose composite. CNTs in the composite structure serve to enhance electron transfer from free-electron products of enzyme oxidative reactions to the electrode surface [20, 21]; Therefore, the composite material strongly contributes to amperometric sensitivity. This material proved to have excellent immobilization of AChE as compared to bovine serum albumin (BSA)-coated gold electrodes. The results showed that AChE was mainly immobilized on the derived oxidized nanocellulose film via carbodiimide crosslinker chemistry (EDC/NHS activation). We compared the performance of the two electrodes on the fluidic chip to investigate enzyme inhibition patterns. Finally, the biosensor was tested for detection of DMT in spiking samples to demonstrate the monitoring of DMT residues in food samples.”
Please refer to lines 14 – 18 of page 2 for your reference.
Reference
- Wani WY, Kandimalla RJL, Sharma DR, Kaushal A, Ruban A, Sunkaria A, Vallamkondu J, Chiarugi A, Reddy PH, Gill KD: Cell cycle activation in p21 dependent pathway: An alternative mechanism of organophosphate induced dopaminergic neurodegeneration. Biochim Biophys Acta Mol Basis Dis 2017, 1863(7):1858-1866.
- Voorhees JR, Remy MT, Erickson CM, Dutca LM, Brat DJ, Pieper AA: Occupational-like organophosphate exposure disrupts microglia and accelerates deficits in a rat model of Alzheimer’s disease. npj Aging and Mechanisms of Disease 2019, 5(1):3.
- Wang A, Cockburn M, Ly TT, Bronstein JM, Ritz B: The association between ambient exposure to organophosphates and Parkinson's disease risk. Occupational and Environmental Medicine 2014, 71(4):275.
- Sánchez-Santed F, Colomina MT, Herrero Hernández E: Organophosphate pesticide exposure and neurodegeneration. Cortex 2016, 74:417-426.
- Judge SJ, Savy CY, Campbell M, Dodds R, Gomes LK, Laws G, Watson A, Blain PG, Morris CM, Gartside SE: Mechanism for the acute effects of organophosphate pesticides on the adult 5-HT system. Chem Biol Interact 2016, 245:82-89.
- Aldewachi H, Al-Zidan RN, Conner MT, Salman MM: High-Throughput Screening Platforms in the Discovery of Novel Drugs for Neurodegenerative Diseases. Bioengineering (Basel) 2021, 8(2).
- Ferreira A, Proença C, Serralheiro ML, Araújo ME: The in vitro screening for acetylcholinesterase inhibition and antioxidant activity of medicinal plants from Portugal. J Ethnopharmacol 2006, 108(1):31-37.
- Pohanka M, Musilek K, Kuca K: Progress of biosensors based on cholinesterase inhibition. Curr Med Chem 2009, 16(14):1790-1798.
- Wang B, Li Y, Hu H, Shu W, Yang L, Zhang J: Acetylcholinesterase electrochemical biosensors with graphene-transition metal carbides nanocomposites modified for detection of organophosphate pesticides. PLOS ONE 2020, 15(4):e0231981.
- Wiesner J, Kříž Z, Kuča K, Jun D, Koča J: Acetylcholinesterases – the structural similarities and differences. Journal of Enzyme Inhibition and Medicinal Chemistry 2007, 22(4):417-424.
- Tota S, Kamat PK, Shukla R, Nath C: Improvement of brain energy metabolism and cholinergic functions contributes to the beneficial effects of silibinin against streptozotocin induced memory impairment. Behavioural Brain Research 2011, 221(1):207-215.
- Scremin OU, Jenden DJ: Chapter 22: Acetylcholine turnover and release: the influence of energy metabolism and systemic choline availability. In: Progress in Brain Research. Edited by Cuello AC, vol. 98: Elsevier; 1993: 191-195.
- Bordone MP, Salman MM, Titus HE, Amini E, Andersen JV, Chakraborti B, Diuba AV, Dubouskaya TG, Ehrke E, Espindola de Freitas A et al: The energetic brain - A review from students to students. J Neurochem 2019, 151(2):139-165.
- Harshit D, Charmy K, Nrupesh P: Organophosphorus pesticides determination by novel HPLC and spectrophotometric method. Food Chemistry 2017, 230:448-453.
- Thompson CM, Prins JM, George KM: Mass spectrometric analyses of organophosphate insecticide oxon protein adducts. Environ Health Perspect 2010, 118(1):11-19.
- Su H, Yeh I-J, Wu Y-H, Jiang Z-H, Shiea J, Lee C-W: Rapid identification of organophosphorus pesticides on contaminated skin and confirmation of adequate decontamination by ambient mass spectrometry in emergency settings. Rapid Communications in Mass Spectrometry 2020, 34(S1):e8562.
- Hua X, Yang J, Wang L, Fang Q, Zhang G, Liu F: Development of an Enzyme Linked Immunosorbent Assay and an Immunochromatographic Assay for Detection of Organophosphorus Pesticides in Different Agricultural Products. PLOS ONE 2013, 7(12):e53099.
- Nair HK, Seravalli J, Arbuckle T, Quinn DM: Molecular recognition in acetylcholinesterase catalysis: free-energy correlations for substrate turnover and inhibition by trifluoro ketone transition-state analogs. Biochemistry 1994, 33(28):8566-8576.
- Nordberg A, Darreh-Shori T, Peskind E, Soininen H, Mousavi M, Eagle G, Lane R: Different cholinesterase inhibitor effects on CSF cholinesterases in Alzheimer patients. Curr Alzheimer Res 2009, 6(1):4-14.
- Liu Y, Zhang J, Cheng Y, Jiang SP: Effect of Carbon Nanotubes on Direct Electron Transfer and Electrocatalytic Activity of Immobilized Glucose Oxidase. ACS Omega 2018, 3(1):667-676.
- Wooten M, Karra S, Zhang M, Gorski W: On the Direct Electron Transfer, Sensing, and Enzyme Activity in the Glucose Oxidase/Carbon Nanotubes System. Analytical Chemistry 2014, 86(1):752-757.

Reviewer 2 Report
Thank you for the opportunity to review this article.
The authors evaluated the topic: A Fluidics-based Biosensor to Detect and Characterize Inhibition Patterns of Organophosphate to Acetylcholinesterase in Food Materials.
The article is well written and I suggest to publish without any modifications.
Author Response
Comment. Thank you for the opportunity to review this article.
The authors evaluated the topic: A Fluidics-based Biosensor to Detect and Characterize Inhibition Patterns of Organophosphate to Acetylcholinesterase in Food Materials.
The article is well written and I suggest to publish without any modifications.
Answer. Thank you for your comments. We have some revisions in the introduction sections to explain the links between organophosphates and neurodegeneration. Please refer to the revised manuscript for your reference.

Reviewer 3 Report
- The quality of the figures are not good, please resubmit the original figures with enough resolution.
- Page 1, line 36: The occurrence of neurodegeneration caused by OP residues and the age composition of the patient population should be briefly described and explained.
- Page 1, line 10:The role of each electrode should be described briefly to help understand how the chip-based biosensor works.
- Page 2, line 23: It is suggested that the ability of DMT to compete for binding sites in the presence of other Ache inhibitors could be investigated at the cellular level.
- Page 2, line 26: The sentence “20 µL of PDDA (20% in water) was added to disperse the CNT-NCT hybrid”, what is the meaning of CNT-NCT hybrid ?
- Page 10, line 17:As for the detection limit, the quantitative limit and the linear range, and error bar should be added.
- Page 10, line 18: The article lacks the comparison of the advantages between the composited electrode and the BSA-coated Au electrode, so the novelty of the composited electrode should be highlighted.
Author Response
Article
A Fluidics-based Biosensor to Detect and Characterize Inhibition Patterns of Organophosphate to Acetylcholinesterase in Food Materials
Dang Song Pham*1, Xuan Anh Nguyen*1, Paul Marsh2, Sung Sik Chu1, Michael P.H. Lau3, Anh H. Nguyen¶2,3, Hung Cao¶1,2,3
1 Biomedical Engineering Department, University of California Irvine, USA
2 Electrical Engineering and Computer Science Department, University of California Irvine, USA
3 Sensoriis, Inc. 7500 212th St SW Ste 208 Edmonds, WA, USA
* Equal contribution
¶ Co-correspondence
1 Affiliation1; [email protected]
2 Affiliation2; [email protected]
* Correspondence: Correspondence: [email protected]; Tel: +1 (949) 824-8478
Hung Cao
Anh H. Nguyen
Electrical Engineering and Computer Sciences
University of California, Irvine, CA 92617
Rebuttal letter
Dear Editors and Reviewers,
We are thankful for the generous comments on the manuscript and we have edited the manuscript to address the concerns.
The reviewers’ comments are in black, while the author's response is in blue, and revised phrases are in red in the revised manuscript.
Review report 3
Comment 1. The quality of the figures is not good, please resubmit the original figures with enough resolution.
Answer. Thank you. All the figures were resubmitted, please refer to the figures in the revised manuscript for your consideration.
Comment 2. Page 1, line 36: The occurrence of neurodegeneration caused by OP residues and the age composition of the patient population should be briefly described and explained.
Answer. Line 36 of page 1 was revised according to the review’s suggestion. Please refer to line 36 of page 36 in the revised manuscript for your consideration. We also quoted them here for your consideration.
“The presence of OP residues in food and dairy products has been reported to cause dopaminergic neurodegeneration [1, 2], Alzheimer's [3], Parkinson's [4], and Amyotrophic Lateral Sclerosis (ALS) diseases [2]. The occurrence of neurodegeneration caused by OPs is associated with acute and chronic effects. The acute effect is associated with AChE inhibition which causes both muscarinic and nicotinic toxicity is due to excessive accumulation of acetylcholine at the neuromuscular junctions and synapses [5]. The chronic effect is due to OPs-induced free radical generation linked with enhanced oxidative stresses that become the key mechanism of their neurotoxic alterations in the long-term effects [6]. While inhibiting AChE activity is the considerable effect for OPs, previous studies found that OPs induce molecular alterations of neuron associated targets, such as hormones [7], neurotransmitters [8], neurotrophic factor [9], and oxidative stress and mitochondrial dysfunction [10] in the chronic effect. Those are increased occurrences of OPs-induced developmental neurotoxicity and the age composition of the patient population [11, 12].”
Comment 3. Page 1, line 10: The role of each electrode should be described briefly to help understand how the chip-based biosensor works.
Answer. Thank you. We added the role of each electrode in the revised manuscript. Please refer to line 7, page 4 for your consideration.
The fluidic channel was manually templated and cut from the six-layer structure to make the fluidic channel. The working electrode (WE), reference electrode (RE), counter electrode (CE), and needles were placed at both ends to connect the device to a micropump. While the reference electrode maintains a constant potential, the working electrode measures the current during the potential scans. The counter electrode directs electricity from the signal source to the working electrode. The second microscope slide was pressed against the parafilm layers, producing the aforementioned sandwich structure. The structure was evenly compressed while the chip was placed in an oven at 45 deg C for 10 minutes; the chip was subsequently taken out and cooled to room temperature (25 deg C). The four sides of the sandwich structure were then sealed by silicone sealant (Grainger, 53DA95).
Comment 4. Page 2, line 23: It is suggested that the ability of DMT to compete for binding sites in the presence of other Ache inhibitors could be investigated at the cellular level.
Answer. Thank you. We agree with you that DMT enables to be used in competitive assays of AChE binding sites in the presence of other AChE inhibitors. The assay can be conducted at the cellular level through measurement of AChE activity in situ or in vivo. However, it is hard to convincible that this sensing platform could differentiate DMT from other inhibitors in binding competition for AChE binding sites. Higher-resolution methods such as isotope/fluorescent labeling for inhibitors could be a solution to measure the competition at the cellular level.
Comment 5. Page 2, line 26: The sentence “20 µL of PDDA (20% in water) was added to disperse the CNT-NCT hybrid”, what is the meaning of CNT-NCT hybrid?
Answer. Thank you for your careful check. This was a typo. The correct phrase is CNT-NC composites instead of CNT-NCT hybrid. Please refer to line 12 of page 3 in the revised manuscript for your consideration.
Comment 6. Page 10, line 17: As for the detection limit, the quantitative limit and the linear range, and error bar should be added.
Answer. We added error bars for Fig. 6 and add the average standard deviation in the text. Please refer to line 17 of page 10 in the revised manuscript for your consideration.
Comment 7. Page 10, line 18: The article lacks the comparison of the advantages between the composited electrode and the BSA-coated Au electrode, so the novelty of the composited electrode should be highlighted.
Answer. We added some descriptions for the novelty of the composited electrode at the line of page 10 in the revised manuscript. Please refer to the manuscript for your consideration.
The LOD values (inset of Figure 6a) were calculated to be 4.1 ± 0.16 nM and 16.9 ± 0.06 nM (in PBS buffer) for the composited electrode (E1) and BSA-coated Au electrode (E2), respectively (Figure 6a, top right), similar to previous reports (Table 1). ). The formation of conductive nanostructures on the surface of the composite electrode provides a favorable microenvironment for AChE activity and increases the electrolysis rate for thiocholine compared to the BSA-coated Au electrode.
Reference
- Wani WY, Kandimalla RJL, Sharma DR, Kaushal A, Ruban A, Sunkaria A, Vallamkondu J, Chiarugi A, Reddy PH, Gill KD: Cell cycle activation in p21 dependent pathway: An alternative mechanism of organophosphate induced dopaminergic neurodegeneration. Biochim Biophys Acta Mol Basis Dis 2017, 1863(7):1858-1866.
- Sánchez-Santed F, Colomina MT, Herrero Hernández E: Organophosphate pesticide exposure and neurodegeneration. Cortex 2016, 74:417-426.
- Voorhees JR, Remy MT, Erickson CM, Dutca LM, Brat DJ, Pieper AA: Occupational-like organophosphate exposure disrupts microglia and accelerates deficits in a rat model of Alzheimer’s disease. npj Aging and Mechanisms of Disease 2019, 5(1):3.
- Wang A, Cockburn M, Ly TT, Bronstein JM, Ritz B: The association between ambient exposure to organophosphates and Parkinson's disease risk. Occupational and Environmental Medicine 2014, 71(4):275.
- Ohbe H, Jo T, Matsui H, Fushimi K, Yasunaga H: Cholinergic Crisis Caused by Cholinesterase Inhibitors: a Retrospective Nationwide Database Study. J Med Toxicol 2018, 14(3):237-241.
- Naughton SX, Terry AV, Jr.: Neurotoxicity in acute and repeated organophosphate exposure. Toxicology 2018, 408:101-112.
- Aguilar-Garduño C, Lacasaña M, Blanco-Muñoz J, Rodríguez-Barranco M, Hernández AF, Bassol S, González-Alzaga B, Cebrián ME: Changes in male hormone profile after occupational organophosphate exposure. A longitudinal study. Toxicology 2013, 307:55-65.
- Slotkin TA, Seidler FJ: Comparative developmental neurotoxicity of organophosphates in vivo: transcriptional responses of pathways for brain cell development, cell signaling, cytotoxicity and neurotransmitter systems. Brain Res Bull 2007, 72(4-6):232-274.
- Dorri SA, Hosseinzadeh H, Abnous K, Hasani FV, Robati RY, Razavi BM: Involvement of brain-derived neurotrophic factor (BDNF) on malathion induced depressive-like behavior in subacute exposure and protective effects of crocin. Iran J Basic Med Sci 2015, 18(10):958-966.
- Farkhondeh T, Mehrpour O, Forouzanfar F, Roshanravan B, Samarghandian S: Oxidative stress and mitochondrial dysfunction in organophosphate pesticide-induced neurotoxicity and its amelioration: a review. Environ Sci Pollut Res Int 2020, 27(20):24799-24814.
- Hung D-Z, Yang H-J, Li Y-F, Lin C-L, Chang S-Y, Sung F-C, Tai SCW: The Long-Term Effects of Organophosphates Poisoning as a Risk Factor of CVDs: A Nationwide Population-Based Cohort Study. PloS one 2015, 10(9):e0137632-e0137632.
- Lin JN, Lin CL, Lin MC, Lai CH, Lin HH, Yang CH, Kao CH: Increased Risk of Dementia in Patients With Acute Organophosphate and Carbamate Poisoning: A Nationwide Population-Based Cohort Study. Medicine (Baltimore) 2015, 94(29):e1187.

Reviewer 4 Report
Please find my suggestions below:
- In section 2.5. Fabrication of chip-based biosensor, it is not clear how the microfluidic chip is formed from Parafilm. Authors neither explained the fabrication steps, nor provided any references. I would like to remind the authors that "One of the most important features of a scientific research paper is that the research must be replicable, which means that the paper gives readers enough detailed information that the research can be repeated". Can the authors provide real pictures/video of the operating chip? Throughout the manuscript, I could not find any real image of the biochip and/or setup.
- Table 1 demonstrates that researchers have developed sensors with lower limit of detection as compared to this work. Can the authors comment on how their sensor has a competitive advantage even with a higher LOD?
Author Response
Article
A Fluidics-based Biosensor to Detect and Characterize Inhibition Patterns of Organophosphate to Acetylcholinesterase in Food Materials
Dang Song Pham*1, Xuan Anh Nguyen*1, Paul Marsh2, Sung Sik Chu1, Michael P.H. Lau3, Anh H. Nguyen¶2,3, Hung Cao¶1,2,3
1 Biomedical Engineering Department, University of California Irvine, USA
2 Electrical Engineering and Computer Science Department, University of California Irvine, USA
3 Sensoriis, Inc. 7500 212th St SW Ste 208 Edmonds, WA, USA
* Equal contribution
¶ Co-correspondence
1 Affiliation1; [email protected]
2 Affiliation2; [email protected]
* Correspondence: Correspondence: [email protected]; Tel: +1 (949) 824-8478
Hung Cao
Anh H. Nguyen
Electrical Engineering and Computer Sciences
University of California, Irvine, CA 92617
Rebuttal letter
Dear Editors and Reviewers,
We are thankful for the generous comments on the manuscript and we have edited the manuscript to address the concerns.
The reviewers’ comments are in black, while the author response is in blue, and revised phrases are in red in the revised manuscript.
Review report 4
Comment 1: In section 2.5. Fabrication of chip-based biosensor, it is not clear how the microfluidic chip is formed from Parafilm. Authors neither explained the fabrication steps, nor provided any references. I would like to remind the authors that "One of the most important features of a scientific research paper is that the research must be replicable, which means that the paper gives readers enough detailed information that the research can be repeated". Can the authors provide real pictures/video of the operating chip? Throughout the manuscript, I could not find any real image of the biochip and/or setup.
Answer. Thank you for your comment. This is considered as a fluidic chip because the diameter of the channel is 200 mm. We added more explanatory text for lines 32-36 of page 3 and line 11 of page 4 in the section of “2.5. Fabrication of chip-based biosensor” for your consideration.
The fluidic chip composed of a six-layer sandwich structure made from laboratory Parafilm (PM-996), with patterned Parafilm layers placed between microscope slides to form an enclosed channel. The channel was prepared by thermally bonding Parafilm and glass slide. The bonding process was performed on a glass slide at 45 oC using a heater plate with the manual press during the bonding process [1]. The bonded layer formed from heated parafilm was cut directly to form the approximately 200-mm desired channels, a thin layer can easily tear off from the glass surface. The process is rapid, cost-effective, and does not require the use of complex procedures to fabricate simple fluidic systems [2]. A basic diagram of the channel structure and preparation steps is shown in Figure 1a and Figure 1c.
The fluidic channel was manually templated and cut from the layer structure to make the fluidic channel. The working electrode (WE), reference electrode (RE), counter electrode (CE), and needles were placed at both ends to connect the device to a micropump. While the reference electrode maintains a constant potential, the working electrode measures the current during the potential scans. The counter electrode directs electricity from the signal source to the working electrode. The second microscope slide was pressed against the parafilm layers, producing the aforementioned sandwich structure. The structure was evenly compressed while the chip was placed in an oven at 60 deg C for 10 minutes; the chip was subsequently taken out and cooled to room temperature (25 deg C). The four sides of the sandwich structure were then sealed by silicone sealant (Grainger, 53DA95) (Figure 1c).
Fig 1a. Schematic of preparation steps for making channels for flow control.
Fig 1b. Real image of the fluidic chip with full-filled spiked milk sample.
Comment 2. Table 1 demonstrates that researchers have developed sensors with a lower limit of detection as compared to this work. Can the authors comment on how their sensor has a competitive advantage even with a higher LOD?
Answer. Thank you. We have some comments and explanations as to how our work could have a competitive advantage compared to previous works in comparison to LOD values. Those are at lines 17-21 of page 10 for your consideration.
The LOD values (inset of Figure 6a) were calculated to be 4.1 ± 0.16 nM and 16.9 ± 0.06 nM (in PBS buffer) for the composited electrode (E1) an BSA-coated Au electrode (E2), respectively (Figure 6a, top right). The formation of conductive nanostructures on the surface of the composite electrode provides a favorable microenvironment for AChE activity and increases the electrolysis rate for thiocholine compared to the BSA-coated Au electrode. Although the LOD value of this work is higher than those of previous reports (Table 1), the biosensor can work well on those fresh vegetable samples which are contaminated by higher 39.2 nM of organophosphate residues (here is for dimethoate) [1]. Moreover, fast, simple, and cost-effective fabrication could be considered as a competitive advantage for mass production.
Reference
- Lu Y, Shi Z, Yu L, Li CM: Fast prototyping of a customized microfluidic device in a non-clean-room setting by cutting and laminating Parafilm®. RSC Advances 2016, 6(88):85468-85472.
- Kim YS, Yang Y, Henry CS: Laminated and infused Parafilm® - paper for paper-based analytical devices. Sens Actuators B Chem 2018, 255(3):3654-3661.
- Yu R, Liu Q, Liu J, Wang Q, Wang Y: Concentrations of organophosphorus pesticides in fresh vegetables and related human health risk assessment in Changchun, Northeast China. Food Control 2016, 60:353-360.

Round 2
Reviewer 1 Report
Dear Editor,
I would like to thank the authors for their efforts to revise the manuscript in the light of the raised concerns and suggestions. All my comments have been addressed by the authors accordingly.
The newly added sections, provided references and enhanced figure have helped towards the improvement of the current version compared to their earlier submission.
I would like to recommend this manuscript for publication.
Best
Author Response
Dear Editors and Reviewers,
We are thankful for the generous comments on the manuscript, and we have edited the manuscript to address the concerns.
The reviewers’ comments are in black, while the author’s response is in blue, and revised phrases are in red in the revised manuscript.
Review report 1
Minor:
Comment 1-Figure 1 is a bit crowded, especially the zoom in part (a). A clearer version will be needed.
Answer. We already fixed the typo-assay and provided a clearer vision for Figure 1. Please refer to Figure 1 in the revised manuscript for your reference.
Comment 2. -P10 line 39: typo- assay.
Answer. We corrected the typo. It was corrected to “with a continuous assay”. Please refer to line 10 of page 10 in the revised manuscript for your reference.

Reviewer 3 Report
The design of the chip lack of the novelty. However, the experimental data is enough with a high quality of presentation.
Author Response
Comment. The design of the chip lack of the novelty. However, the experimental data is enough with a high quality of presentation.
Answer. Thank you for your valuable comments. We agree that the device is not an innovative design because we forwarded it to simple design and operations. In the future, we will add more electronic innovations in the device such as wireless and cloud system to enhance its design and performance. Thank you.
Reviewer 4 Report
Thanks for the revision. There are still some discrepancies that need to be addressed before the paper is published. Please check below.
- In page 6, line #9, the scan rate is mentioned to be 0.2 mV/s. However, in the caption of Figure 4, the scan rate was used to be 0.2 V/s. Which one is correct?
- Thanks for adding Figure 1c. However, it is still not clear how the WE, CE and RE can be just placed that way. Aren't they made on a Si/SiO2 substrate? Are you directly placing the substrate then which carries the electrodes? What is the substrate thickness? Won't that increase the channel depth? How are you cutting parafilm manually to make it exactly 200 micrometers deep?
- The plots in Figure 4a and 4b are for different concentrations. What do those different concentrations represent?
Author Response
Dear Editors and Reviewers,
We are thankful for the generous comments on the manuscript, and we have edited the manuscript to address the concerns.
The reviewers’ comments are in black, while the author’s response is in blue, and revised phrases are in red in the revised manuscript.
Review report 4
Thanks for the revision. There are still some discrepancies that need to be addressed before the paper is published. Please check below.
Comment 1.
In page 6, line #9, the scan rate is mentioned to be 0.2 mV/s. However, in the caption of Figure 4, the scan rate was used to be 0.2 V/s. Which one is correct?
Answer. Thank you for your careful check. We used a scan rate of 0.2 mV/s. We corrected the unit of V/s in the caption of Figure 4, Figure 4C to the unit of mV/s. Please refer to line 9, page 6, and Figure 4 for your reference.
Comment 2.
1. Thanks for adding Figure 1c. However, it is still not clear how the WE, CE and RE can be just placed that way.
Answer. To place the electrodes in their locations, we did it manually. First, we did step by step as shown in Figure 1c. Before doing the fabrication, each parafilm layer has its thickness of ~127 -130 µm, thus 6 layers x 130 µm ~780 µm. The total thickness was reduced due to stretching and compression to reach approximately 200 µm.
- Aren't they made on a Si/SiO2 substrate?
Answer. Si/SiO2 was used as the substrate to make gold electrodes, used as working electrodes accordingly. The fluidic chip was made from a glass slide and six parafilm layers. The working, reference and counter electrode were placed in the fluidic chip manually.
- Are you directly placing the substrate then which carries the electrodes?
Answer. The fluidic chip includes two pieces of glass slides. As shown in Figure 1c, the first glass slide was used as the substrate to fabricate the fluidic channel which was made from six layers of parafilm. After shaping the fluidic system, reaction chamber, and three grooves for three electrodes, respectively. We inserted the electrodes into their grooves and closed the fluidic chip with the second slide, used silicon to seal the chip.
- What is the substrate thickness? How are you cutting parafilm manually to make it exactly 200 micrometers deep?
Answer. Keeping this order and decreasing resistance between RE and CE or directly diminishing the distance those electrodes are to mitigate an ohmic drop event [1]. For this project, gold electrodes (1×1 mm2 square) (WE) were fabricated by e-beam evaporation of 20 nm Cr and 200 nm gold (Au) onto SiO2/Si substrates. Ag/AgCl and Pt wire were used as a RE and CE.
The first composite electrode was prepared by casting 5 µL of the CNT-PDDA-NC suspension onto the surface of the bare gold electrode and placing the electrode in a vacuum desiccator to evaporate the solvent. As controls, bare gold electrodes were treated with 0.5 mM 11-Mercaptoundecanoic acid (11-MUA) and coated by 10 L of AChE: BSA (ratio 1:1 v/v). The total thickness of the gold electrode (~100 microns) and substrate (~40 microns) is less than the height of 6 parafilm layers, about 200 microns after compression and stretching by the heater at 60oC for 10 mins. Thus, the height of WE with substrate won’t affect the channel depth. A frame of channel system drawn on a top layer of the 6 parafilm layers on a microscope slide is cut by a surgical knife. The thickness of those parafilm layers after compression and oven is about 200 microns.
- Won't that increase the channel depth?
Answer. We cut the parafilm layers to form three grooves corresponding to three electrodes, respectively. So, the channel depth will not increase. Please see the model below for your reference.
Comment 3.
The plots in Figure 4a and 4b are for different concentrations. What do those different concentrations represent?
Answer.
We use the same concentrations of 0.01 – 50 M of ATC in 10 mM phosphate buffer supported with 0.1 M KCl for both experiments on composited electrodes (Fig. 4a) and BSA-coated gold electrodes (Fig. 4b). The goal of these experiments is to detect the Limit of Detection (LOD) and to compare the sensitivity of composited electrodes and BSA-coated gold electrodes for detection of DMT. As we mentioned on page 6, from line #4 to line #26, AChE coated on electrode surface specifically catalyzes ATC substrates and produces thiocholine that changes the current. Figure 4a and Figure 4b showed that the AChE catalytic activity on the active surface area of the composite electrode stronger than BSA-coated gold electrodes. We added one decimal (for example 50.0 µM) in the caption of Figure 1a, please refer to Figure 4c in the revised manuscript for your consideration.
Reference.
- Elgrishi, N., Rountree, K. J., McCarthy, B. D., Rountree, E. S., Eisenhart, T. T., & Dempsey, J. L. (2017). A practical beginner’s guide to cyclic voltammetry. Journal of Chemical Education, 95(2), 197-206. doi: 10.1021/acs.jchemed.7b00361
